# TAD-Net: Reinforced Anomaly Generation and Wavelet-enhanced Prediction for Temporal Anomaly Detection

## Abstract

In dynamic graph environments, structure-based anomaly detection is essential for applications such as identifying fraudulent calls, fake accounts, and social bots. While existing methods typically monitor changes in structural features to detect anomalies, they often fail to account for concept drift—where natural, gradual changes in network structure are incorrectly flagged as anomalies. To address this limitation, we introduce **T**emporal **A**nomaly **D**etection **NET**work (TAD-Net), a framework specifically designed to reduce the impact of concept drift and improve anomalous node detection. TAD-Net consists of three main components: (i) temporal feature extractor; (ii) reinforced anomaly generator; and (iii) wavelet-enhanced fusion predictor. The temporal feature extractor identifies changes in node features via dynamic behavior projection, distinguishing between normal network evolution and true anomalies. Working in tandem with the anomaly detector, it leverages structural-difference attention to learn robust representations for abnormal node detection. To address limited labeled anomalies, the reinforced anomaly augmenter generates synthetic anomalous samples using reinforced generative adversarial networks. The wavelet-enhanced fusion predictor improves adaptability to structural changes by integrating high-frequency features, maintaining anomaly sensitivity as the network evolves. Experiments on real-world datasets show that TAD-Net outperforms state-of-the-art methods, achieving over 6% AUC improvement under concept drift. The code is available at https://anonymous.4open.science/r/TAD-Net-B26A.

## 1 Introduction

Many real-world systems—such as financial transaction networks (Choi et al., 2019), social media platforms (Mancino et al., 2025), and internet communication infrastructures (Al-Heety et al., 2025)—are inherently dynamic, with structures and interactions that evolve over time. These systems are commonly represented as dynamic graphs to capture their temporal evolution. Within such networks, certain nodes may exhibit behaviors that deviate markedly from the norm; these anomalous nodes can disrupt normal operations and compromise user security. For example, in financial networks, fraudsters exploit system vulnerabilities to conduct illicit transactions. On social platforms, malicious bots disseminate misinformation and generate fake engagement. In internet communications, cyberattacks can result in privacy breaches and substantial financial losses. Consequently, robust anomaly detection methods are essential to identify and mitigate the risks posed by anomalous nodes in dynamic network environments.

As illustrated in Figure 1, dynamic graphs are subject to concept drift, where natural and expected changes in node behavior or network structure are mistakenly flagged as anomalies. For example, a classifier trained on transaction frequencies from the previous week may fail to detect current fraudulent activity in a financial network. On special occasions such as shopping days, a surge in transaction frequency can cause regular users to be misclassified as fraudsters, simply because the model has not adapted to the new distribution. Recent works such as Hong et al. (2025) have combined generative adversarial mechanisms with meta-learning to synthesize additional anomalies and facilitate rapid adaptation. These approaches indeed enrich the training space and enhance robustness under varying conditions. However, conventional GAN-based generators inherently produce anomalies that remain close to the observed training distribution. As a result, they are ineffective in

capturing emerging or previously unseen anomalies induced by concept drift, where the underlying graph distribution evolves over time.

Addressing concept drift in dynamic graph anomaly detection presents two primary challenges. _First_, anomaly samples are inherently scarce, as anomalous behaviors are rare and labeled data is extremely limited Liu et al. (2025); Ma et al. (2023). In dynamic graphs, nodes and edges evolve over time, and anomalies may appear only transiently or in subtle forms, making it difficult for models to capture diverse anomaly patterns from limited observations. Generative approaches, such as GAN-based mechanisms, have been explored to mitigate this scarcity by synthesizing additional anomalies. However, conventional generators inherently produce samples that remain close to the observed training distribution, rendering them ineffective in capturing emerging or previously unseen anomalies induced by concept drift, where the underlying graph distribution evolves over time. This limitation highlights the necessity for a generation mechanism that

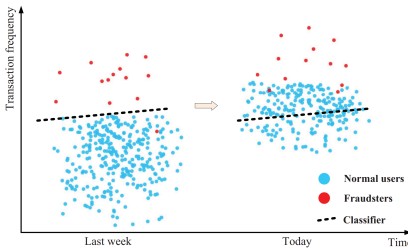

Figure 1: This figure illustrates a financial network's fraud detection using transaction frequency from last week to today. The dashed black line represents the anomaly classifier trained on the graph data from last week.

not only alleviates sample scarcity but also adapts to temporal dynamics, thereby producing anomalies consistent with evolving graph patterns. _Second_, dynamic graphs are subject to concept drift, where natural changes in node behavior and network structure may resemble anomalous patterns, making them difficult to distinguish. While intuitively, abrupt or localized anomalies often correspond to high-frequency components in the graph spectrum, natural evolution tends to be smooth and concentrated in low-frequency bands Ortega et al. (2018). Existing graph neural networks inherently perform low-pass filtering Zhang et al. (2025), which suppresses these high-frequency components and diminishes sensitivity to subtle anomalies. Furthermore, prior wavelet-based methods Lu & Ghorbani (2008); Donnat et al. (2018) have not been systematically integrated with temporal modeling to explicitly separate anomalies from natural evolution. These observations highlight the necessity of a framework that can preserve high-frequency anomaly-relevant information while adapting to evolving network dynamics under concept drift.

To address these challenges, we propose **T**emporal **A**nomaly **D**etection **NET**work (TAD-NET), a novel framework designed to detect anomalies in dynamic graphs under concept drift and limited anomaly samples. TAD-NET comprises three complementary modules: (i) a **projection-based temporal feature extractor** that captures relative changes in node features over time, helping to disentangle natural evolution from anomalous deviations and providing meaningful inputs for subsequent anomaly generation; (ii) a **reinforced anomaly generator**, which integrates generative adversarial networks with reinforcement learning principles to synthesize realistic anomaly samples. Here, the generator acts as the _agent_, the discriminator serves as the _environment_, the generated features form the _state_, and the discriminator's output probability serves as the _reward_, guiding the generator to produce high-quality anomalies consistent with temporal dynamics. The reinforcement learning framework, including discount factors and separate optimizers for generator and discriminator, ensures effective exploration and stable adversarial training; (iii) a **wavelet-enhanced fusion predictor** that explicitly preserves high-frequency signals in node features, allowing the model to distinguish abrupt, anomalous changes from smooth concept drift in evolving graph structures. By jointly leveraging temporal feature extraction, reinforced anomaly generation, and high-frequency signal preservation, TAD-NET effectively mitigates sample scarcity, maintains sensitivity to subtle anomalies, and adapts robustly to dynamic graph evolution. We summarize our key contributions as follows:

- We introduce TAD-NET, a modular framework for dynamic graph anomaly detection that explicitly addresses concept drift while preserving sensitivity to genuine anomalies.
- We design a reinforced anomaly generator combining adversarial learning with reinforcement learning principles to synthesize realistic anomalies under evolving graph conditions, addressing the scarcity of labeled anomaly samples.
- We incorporate a wavelet-enhanced fusion predictor to capture high-frequency structural and feature changes, allowing robust separation of abrupt anomalies from smooth temporal evolution.

- Extensive experiments on multiple real-world dynamic graph datasets demonstrate that TAD-NET consistently outperforms state-of-the-art baselines, validating the effectiveness of our integrated approach.

## 2 RELATED WORK

**Anomaly node detection in dynamic networks.** Dynamic graph anomaly detection has focused on structural changes in networks. Methods like NetWalk Yu et al. (2018) use random walks and autoencoders to detect anomalies through node clustering. NFGCN Wang et al. (2022) and STGCNs Mu et al. (2022) apply GCNs to capture both spatial and temporal dependencies, aiding detection in recommender systems and video segments.

Other methods include TBCCA Zhang et al. (2023), which detects fraud by modeling temporal and structural dependencies, and JODIE Kumar et al. (2019), which predicts anomalies by updating node embeddings via recurrent networks. APAN Wang et al. (2021a) enables real-time anomaly detection by decoupling graph computation from inference, while TGAT Xu et al. (2020b) uses self-attention for temporal edge information. Recent works like GDN Ding et al. (2021a) enhance anomaly detection using minimal labeled data, and SAD Tian et al. (2023) integrates memory and pseudo-label contrastive learning for better performance on large unlabeled datasets. Despite advancements, these methods face challenges with concept drift, where natural changes in network structure may be misidentified as anomalies, highlighting the need for more robust methods.

**High-Frequency Feature Processing with Discrete Wavelet Transform.** Recent work has shown that high-frequency feature extraction using DWT is highly effective for anomaly detection in dynamic environments, such as financial networks (Wang et al., 2021b) and IoT systems. By applying DWT, models can isolate fine-grained frequency components that are often indicative of anomalous behavior. For instance, EawT (Zhou et al., 2020) combines wavelet transforms with convolutional operations and introduces a wavelet-based loss to refine feature representations for anomaly detection. DWT is also computationally efficient, making it suitable for real-time and resource-constrained scenarios (Li et al., 2022). Other approaches, such as MWNet (Shang et al., 2024) and Meta-MWDG (Xie et al., 2024), further leverage DWT to model frequency differences and capture both frequency-domain and temporal dependencies. AutoWave (Liu et al., 2020) uses autoencoders with DWT to reconstruct time series in both time and frequency domains, improving sequence anomaly detection. Collectively, these studies demonstrate the versatility and effectiveness of DWT-based methods for robust anomaly detection.

## 3 PRELIMINARIES

In this section, we introduce the notations and problem definition for dynamic graph anomaly detection. A quick background on the core concepts is provided in the Appendix B.

**Notations.** We represent a dynamic graph as $\mathcal{G} = (\mathcal{V}, \mathcal{E}, \mathcal{X})$, where $\mathcal{V}$ denotes the set of nodes $\mathcal{E}$ denotes the set of temporal edges and $\mathcal{X} \in \mathbb{R}^{\mathcal{V} \times d}$ denotes the set of node features. Each edge $e_i = (v_i, v_j, t_i) \in \mathcal{E}$ indicates an interaction or event from node $v_i$ to node $v_j$ at time $t_i$. The set of temporal edges is denoted as $\mathcal{E} = \{e_1, e_2, \ldots, e_m\}$, where $m$ is the total number of temporal edges. We partition the dynamic graph $\mathcal{G}$ into two subgraphs based on the time dimension: the historical graph $\mathcal{G}_{\text{history}}$ (containing earlier interactions) and the newly emerged graph $\mathcal{G}_{\text{new}}$ (containing recent interactions). For each node $v_i$ at time $t$, we define its anomaly label as $y_i^t$, where $y_i^t = 0$ indicates a normal node and $y_i^t = 1$ indicates an anamolous node. We also present list of widely used notations in Appendix (Table 3).

**Problem Definition.** Given a historical subgraph $\mathcal{G}_{\text{history}}$ (with many labeled nodes) and a new subgraph $\mathcal{G}_{\text{new}}$ (with few labeled nodes), the objective is to detect anomalous nodes in $\mathcal{G}_{\text{new}}$ in the presence of concept drift—that is, when the underlying graph structure and data distribution evolve over time, reducing the reliability of models trained solely on historical data. Formally, let $\mathcal{G}_{\text{new}} = (\mathcal{V}_{\text{new}}, \mathcal{E}_{\text{new}})$, where each node $v_i \in \mathcal{V}_{\text{new}}$ is associated with a feature vector $\mathbf{x}_i$ and an anomaly label $y_i \in \{0, 1\}$. The goal is to learn a function $f$ that, for each $v_i$, predicts $\hat{y}_i = f(\mathbf{x}_i, \mathcal{G}_{\text{new}})$, assigning a label of normal (0) or anomalous (1) to each node in the new subgraph.

## 4 TAD-NET

**Overview.** In this section, we introduce **T**emporal **A**nomaly **D**etection **NET**work (TAD-NET), an end-to-end framework designed to improve anomaly detection in dynamic graphs under concept

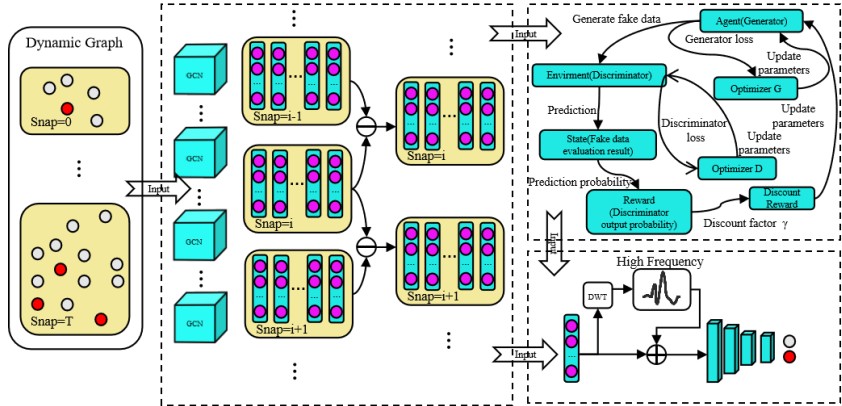

Figure 2: The TAD-NET framework integrates: dynamic behavior projection to quantify node feature differences across temporal states, reinforced adversarial training for realistic anomaly synthesis, and wavelet-based high-frequency analysis to sustain sensitivity under structural evolution. This unified architecture enables robust anomaly detection in concept drift scenarios through coordinated feature learning and distribution adaptation.

drift. As illustrated in Figure 2, TAD-NET is composed of three key modules: (i) Temporal Feature Extractor, which captures evolving node characteristics by modeling temporal feature changes; (ii) Reinforced Anomaly Generator, which leverages generative and reinforcement learning techniques to synthesize diverse anomalous samples and address the scarcity of labeled anomalies; and (iii) Wavelet-Enhanced Fusion Predictor, which integrates high-frequency features via wavelet transform to enhance sensitivity to subtle anomalies amid structural changes. Together, these modules enable TAD-NET to effectively differentiate genuine anomalies from normal patterns of network evolution, ensuring robust performance in dynamic environments.

## 4.1 TEMPORAL FEATURE EXTRACTOR

The temporal feature extractor module is designed to capture evolving patterns in dynamic graphs by processing a sequence of graph snapshots $G_t$ and their associated node features $X_t \in \mathbb{R}^{N_t \times d}$, where $N_t$ denotes the number of nodes at time $t$ and $d$ is the feature dimension. The temporal feature extractor outputs two key matrices: the node embedding matrix $H_t \in \mathbb{R}^{N_t \times d'}$, and the temporal change matrix $\Delta H_t \in \mathbb{R}^{N_t \times d'}$, which quantifies feature variations across consecutive time steps.

At the first time step ($t = 1$), node embeddings are initialized using a graph convolutional network Zhang et al. (2019), as follows:

$$H_1 = \sigma \left( \tilde{D}^{-1/2} \tilde{A} \tilde{D}^{-1/2} X_1 W \right) \tag{1}$$

where $\tilde{A} = A + I$ is the adjacency matrix with self-loops, $\tilde{D}$ is its degree matrix, $W$ is a learnable weight matrix, and $\sigma$ is a nonlinear activation function. For each subsequent time step ($t > 1$), the feature matrix $X_t$ is updated by replacing the first $N_{t-1}$ rows with the previous embeddings $H_{t-1}$:

$$X_t[0 : N_{t-1}] = H_{t-1} \tag{2}$$

The updated features are then passed through the graph convolutions to obtain the current embeddings:

$$H_t = \sigma \left( \tilde{D}^{-1/2} \tilde{A} \tilde{D}^{-1/2} X_t W \right) \tag{3}$$

To capture temporal dynamics, the difference between embeddings at consecutive time steps is computed. Specifically, we unify the two cases of node number variation into a single definition:

$$\Delta H_t = \begin{cases} H_t - H_{t-1}[0{:}N_t, :], & N_t \leq N_{t-1}, \\ H_t - \texttt{Pad}(H_{t-1}, N_t), & N_t > N_{t-1}, \end{cases} \tag{4}$$

where $\texttt{Pad}(H_{t-1}, N_t)$ pads $H_{t-1}$ (e.g., with zeros) to have $N_t$ rows. This unified formulation eliminates the need to separately write equations (5) and (6). This mechanism ensures that $\Delta H_t$ accurately reflects temporal changes in the network, even as the node set evolves, thereby providing rich temporal representations for downstream anomaly detection.

**Theoretical Motivation.** When the graph-derived representation evolves smoothly over time, the temporal difference of node embeddings $\Delta H_t$ defined in Eq. equation 4 remains within a predictable range. In contrast, anomalies introduce abrupt changes in the inputs, which propagate through the encoder and manifest as larger $\|\Delta H_t\|_F$. Therefore, temporal differencing naturally amplifies anomalies, which underpins the design of our *Temporal Feature Extractor*. Let $X_t \in \mathbb{R}^{N_t \times d}$ be the node-feature matrix at time $t$, and let $H_t = f(A, X_t) \in \mathbb{R}^{N_t \times h}$ denote the output of one graph-convolution layer applied to $X_t$ with a (possibly self-loop augmented) adjacency $A$ fixed at time $t$ (extensions to time-varying $A_t$ are discussed in the remarks). Based on Eq. equation 4, the temporal difference $\Delta H_t$ directly captures variations in representations while adapting to dynamic node sets, thus providing a mathematically consistent basis for subsequent anomaly detection.

**Assumption 4.1** (Lipschitz temporal evolution and non-expansive padding). *For* normal *evolution the feature sequence is $L_X$-Lipschitz in time: $\|X_t - X_{t-1}\|_F \leq L_X$. Let one GCN layer be $f(A, X) = \sigma(\tilde{D}^{-1/2} \tilde{A} \tilde{D}^{-1/2} XW)$, where $\sigma$ is $L_\sigma$-Lipschitz, $\|\tilde{D}^{-1/2} \tilde{A} \tilde{D}^{-1/2}\|_2 \leq L_A$, and $W$ is a trainable weight matrix. The padding operator is non-expansive: for any $U, V$ and any $n$, $\|Pad(U, n) - Pad(V, n)\|_F \leq \|U - V\|_F$.*

**Lemma 4.1** (Stability of one-step embedding). *Under the above assumption, the mapping $X \mapsto H = f(A, X)$ is $L_f$-Lipschitz in Frobenius norm with $L_f \leq L_\sigma L_A \|W\|_2$. That is, for any $X, X'$,*

$$\|f(A, X) - f(A, X')\|_F \leq L_f \|X - X'\|_F.$$

**Theorem 4.1** (Detection margin under anomaly perturbation). *Suppose an anomaly increases the input temporal jump by at least $\delta > 0$, i.e., $\|X_t - X_{t-1}\|_F \geq L_X + \delta$. If, moreover, the encoder satisfies the local gain condition in Assumption F.2, then $\|\Delta H_t\|_F \geq \mu_f (L_X + \delta) - R_t$. Therefore the excess over the normal bound $\tau_t = L_f L_X + R_t$ obeys $\|\Delta H_t\|_F - \tau_t \geq \mu_f \delta - (L_f - \mu_f) L_X - 2R_t$. In particular, a sufficient condition for a positive detection margin is: $\mu_f \delta > (L_f - \mu_f) L_X + 2R_t$.*

Corresponding proofs and in-depth analyses are provided in Appendix F.1.

## 4.2 REINFORCED ANOMALY GENERATOR

**Theoretical Motivation.** Standard GAN-based anomaly synthesis often suffers from mode collapse, concentrating on high-density regions of the empirical anomaly distribution. To address this, we integrate reinforcement learning: the generator explores the anomaly feature space, while the discriminator provides a reward encouraging both realism and diversity. The generator is trained with policy-gradient updates (Lemma F.2, Appendix F.2) and entropy-regularized rewards, promoting exploration of low-density regions. Theoretical results (Theorems F.2 and F.3, Appendix F.2) ensure non-zero probability for all data modes and allow support expansion beyond the observed anomalies, enabling adaptation to unseen patterns. Detailed proofs are in the appendix F.2.

**Generator-Discriminator Interaction.** At each time step $t$, the reinforced anomaly generator receives the anomaly-related feature matrix $\Delta H_t \in \mathbb{R}^{N_t \times d'}$ and selects a subset of anomalies $X_a \in \mathbb{R}^{M \times d}$. The generator $G$ produces synthetic anomalies $\hat{X}_a = G(Z)$, with noise $Z \sim P_z$, while the discriminator $D$ evaluates both real and generated anomalies. The reward from $D$ forms a reinforcement learning loop, naturally integrating the theoretical guarantees mentioned above.

**Training Objective and Parameter Updates.** The discriminator loss is

$$L_D = -\mathbb{E}_{X_a \sim P_{\text{data}}}[\log D(X_a)] - \mathbb{E}_{Z \sim P_z}[\log(1 - D(G(Z)))] \quad (5)$$

The generator loss incorporates adversarial and reward terms:

$$L_G = -\mathbb{E}_{Z \sim P_z}[\log D(G(Z))] + \gamma \log D(G(Z)) \quad (6)$$

Parameters are updated via gradient descent:

$$\theta_d \leftarrow \theta_d - \eta_d \nabla_{\theta_d} L_D \quad (7)$$

$$\theta_g \leftarrow \theta_g - \eta_g \nabla_{\theta_g} L_G \quad (8)$$

This strategy allows the generator to explore underrepresented anomaly regions while maintaining realism, with theoretical backing ensuring coverage and adaptation to evolving anomalies (see Appendix F.2 for detailed lemmas, theorems, and proofs).

### 4.3 Wavelet-enhanced Fusion Predictor

The wavelet-enhanced fusion predictor improves anomaly detection by explicitly capturing high-frequency deviations in node features (Lemma F.4, Proposition F.3). While concept drift in dynamic graphs typically manifests as smooth, low-frequency changes, true anomalies induce abrupt, localized deviations Lu & Ghorbani (2008); Iqbal et al. (2025). By leveraging the discrete wavelet transform (DWT) to separate high- and low-frequency components, the module enables more robust detection of genuine anomalies (Theorem F.4); see Appendix F.3 for detailed theoretical justification.

For each feature vector $v$ in the temporal difference set $\Delta H_t$ or the synthetic anomalies $\hat{X}_a$, DWT decomposes $v$ into low- and high-frequency components $\texttt{C} = \text{DWT}(v)$, and the high-frequency component is extracted as $H_{\text{high}}^v = \texttt{C}[1]$. These are fused with the original features via a weighted sum:

$$H_{\text{fusion}}^v = v + \alpha H_{\text{high}}^v, \tag{9}$$

where $\alpha$ controls the influence of high-frequency components. The fused feature is then input to a neural network classifier, trained with cross-entropy loss:

$$L_{\text{CE}} = - \sum_{v \in H_t \cup X_a} [y_v \log(\hat{y}_v) + (1 - y_v) \log(1 - \hat{y}_v)]. \tag{10}$$

**Theoretical Motivation.** The high-frequency fusion in Eq. 9 amplifies anomaly-induced deviations relative to smooth temporal evolution. Intuitively, smooth concept-drift changes lie in low-frequency DWT coefficients, while abrupt anomalies appear in high-frequency components. Formally, if $v = s_t + a_t$ with $s_t$ smooth and $a_t$ anomalous, the fused feature satisfies

$$H_{\text{fusion}}^v = s_t + (1 + \alpha)a_t,$$

which increases the signal-to-noise ratio of anomalies:

$$\frac{\|(1 + \alpha)a_t\|_2}{\|s_t\|_2} > \frac{\|a_t\|_2}{\|s_t\|_2}.$$

This justifies the design choice. Rigorous derivations and proofs are provided in Appendix F.3.

### 4.4 Model Training

Training of TAD-NET follows a three-phase procedure coordinating temporal feature extraction, reinforced anomaly generation, and wavelet-enhanced prediction, operating on dynamic graph snapshots $\{G_t\}_{t=1}^T$ with node features $\{X_t\}_{t=1}^T$. Separate learning rates $\eta_d$, $\eta_g$, and $\eta$ are used for the discriminator, generator, and predictor, respectively. A detailed algorithm is provided in Appendix D (Algorithm 1).

**Phase 1: Temporal Feature Extraction.** Node embeddings $H_t$ are updated using the graph convolution (Eq. 3), and temporal differences $\Delta H_t$ are computed with the padding/truncation mechanism (Eq. 4). This ensures that the embeddings capture evolving patterns while handling dynamic node sets.

**Phase 2: Reinforced Anomaly Generation.** The generator produces synthetic anomalies $\hat{X}_a$ from noise inputs, and the discriminator evaluates real versus generated anomalies. Training follows the adversarial-reinforcement framework, optimizing $L_D$ and $L_G$ (Eqs. 5–6) with alternating updates of parameters $\theta_d$ and $\theta_g$ (Eqs. 7–8).

**Phase 3: Wavelet-Enhanced Prediction.** The predictor fuses the original features with high-frequency components (Eq. 9) and is trained using the cross-entropy loss $L_{\text{CE}}$ (Eq. 10). This amplifies subtle anomaly signals while remaining robust to low-frequency, concept-drift-induced changes.

## 5 Experimentation

In this section, we evaluate the performance of our proposed framework on three real-world social media datasets. We first introduce the datasets and the experimental settings. Then, we present the results and discuss the performance of our framework. Owing to lack of space, we report additional results in the Appendix H.

Table 1: Statistics of the real-world datasets

| Datasets | Nodes | Edges | Anomalies | Timespan |
|---|---|---|---|---|
| Wikipedia | 9,227 | 157,474 | 217 | 30 days |
| Reddit | 10,984 | 672,447 | 366 | 30 days |
| Mooc | 7,074 | 333,734 | 4,066 | 30 days |

## 5.1 EXPERIMENTAL SETTINGS

**Datasets.** We conduct experiments on three widely used real-world social media datasets: (i) Wikipedia Wang et al. (2020), (ii) Reddit Nguyen et al. (2020), and (iii) Mooc Toghani et al. (2022), each exhibiting unique structural properties and anomaly patterns. For all datasets, we follow a consistent data split: 70% for training, 10% for validation, and 20% for testing. To capture temporal dynamics, we extract 5 network snapshots per dataset based on their respective timestamps. Key dataset statistics are presented in Table 1, with additional details available in Appendix G.2.

**Baselines.** To evaluate the performance of TAD-NET we use following state-of-the-art methods as baselines: (i) TGAT Xu et al. (2020a), (ii) GDN Ding et al. (2021b), (iii) SAD Tian et al. (2023), (iv) TADDY Liu et al. (2021), (v) MAMF Hong et al. (2025). Further details about the baselines are provided in Appendix G.3.

**Evaluation Metrics.** We assess model performance using AUC-ROC, Precision, F1-Score and AUPR, which are standard metrics for anomaly detection in dynamic graphs. We use AUC as the primary metric following baseline comparisons Xu et al. (2020a); Tian et al. (2023), with other metrics providing complementary analysis. Further details and mathematical formulation of these metrics are detailed in Appendix G.4.

**Experimental Setup.** For TAD-NET, the node embedding dimension in the temporal feature extractor is set to $k = 128$. The reinforced anomaly generator synthesizes realistic anomalous features from training snapshots to augment data and improve detection. The wavelet-enhanced fusion predictor uses a four-layer MLP to extract features effectively while mitigating overfitting. All models, including TADNet and baselines, are trained for 100 epochs with a learning rate of $5 \times 10^{-5}$. Baselines follow the hyperparameters reported in their original papers. Each experiment is repeated 20 times to ensure statistical robustness.

## 5.2 MAIN RESULTS

The performance comparison between our method, TAD-NET, and baseline models is shown in Table 2. TAD-NET consistently outperforms all baselines across datasets and metrics. On Wikipedia, it achieves 97.87% AUC (6.66% higher than MAMF), 90.11% precision, 90.10% F1, and 83.75% AUPR. Similar gains appear on Reddit and Mooc, especially in F1 and AUPR.

Table 2: Performance comparisons of different methods on all datasets in terms of AUC (%), Precision (%), F1 (%), and AUPR (%). Bold values indicate the best performance.

| Method | Wikipedia | | | | Reddit | | | | Mooc | | | |
|---|---|---|---|---|---|---|---|---|---|---|---|---|
| | AUC | Precision | F1 | AUPR | AUC | Precision | F1 | AUPR | AUC | Precision | F1 | AUPR |
| TGAT | 83.23 | 1.84 | 1.05 | 0.92 | 67.06 | 3.23 | 0.95 | 0.33 | 66.88 | 6.23 | 2.01 | 1.11 |
| GDN | 85.12 | 6.78 | 3.90 | 1.78 | 67.02 | 0.75 | 0.49 | 0.16 | 66.21 | 3.86 | 2.65 | 3.28 |
| SAD | 86.77 | 1.67 | 4.26 | 1.98 | 68.77 | 0.16 | 0.59 | 0.23 | 69.44 | 3.29 | 2.34 | 2.81 |
| TADDY | 84.72 | 8.31 | 15.30 | 8.72 | 67.95 | 8.16 | 15.00 | 8.06 | 68.47 | 10.97 | 19.74 | 11.17 |
| MAMF | 91.21 | 89.36 | 77.31 | 69.83 | 71.35 | 56.59 | 65.67 | 61.88 | 75.64 | 78.42 | 53.48 | 55.69 |
| TAD-NET | **97.87** | **90.11** | **90.10** | **83.75** | **93.31** | **89.95** | **83.19** | **83.75** | **81.41** | **82.32** | **74.39** | **66.64** |

Baselines like TGAT, GDN, and SAD face clear limitations. TGAT's self-attention misses subtle behaviors, resulting in low AUC and F1. GDN struggles with complex temporal variations despite labeled anomalies, causing low precision and AUPR. SAD's pseudo-label contrastive learning is less effective on imbalanced datasets like Reddit. TADDY's single-transformer limits behavior diversity modeling. MAMF uses GAN-generated anomalies but lacks effective high-frequency feature extraction, reducing performance on complex data.

TAD-NET's strength lies in integrating multi-scale feature fusion, attention mechanisms, DWT-based high-frequency extraction, and adversarial sample generation via GANs and reinforcement learning. This combination captures fine details, isolates key behaviors, emphasizes transient signals, and enriches training data, enhancing robustness and generalization.

## 5.3 ABLATION STUDIES

To evaluate the contribution of key components in TAD-NET, we perform ablation studies by selectively removing specific modules. The goal is to quantify the impact of each module on the model's AUC, a key metric for distinguishing normal and anomalous instances in dynamic graphs. We investigate three configurations:

- TAD-NET(–H): removing the high-frequency feature amplification.
- TAD-NET(–R): removing the reinforcement learning rewards mechanism.
- TAD-NET(–B): removing both high-frequency amplification and reinforcement learning.

Figure 3 presents the AUC results for each ablation configuration. The full version of TAD-NET, which includes both high-frequency feature amplification and reinforcement learning rewards, achieves the highest AUC across all datasets. This demonstrates that the combination of these two components enables the model to capture subtle, fine-grained anomalies while adapting to evolving data distributions in dynamic graphs.

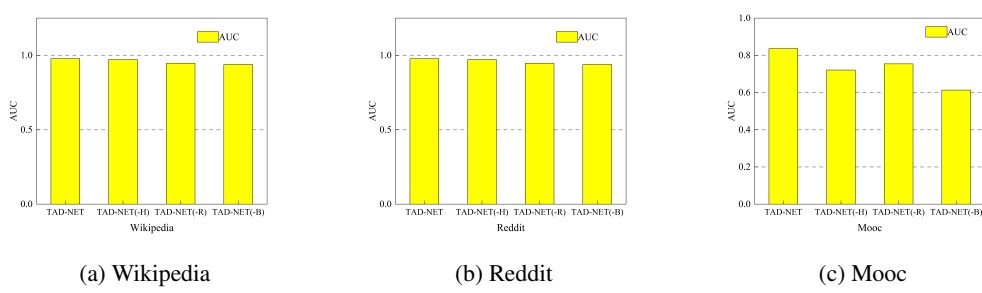

| (a) Wikipedia | (b) Reddit | (c) Mooc |

Figure 3: AUC values of ablation study on different datasets.

To isolate the effect of high-frequency features, we remove the high-frequency amplification module, *i.e.,* TAD-NET(–H). This leads to a substantial decrease in AUC on all datasets, confirming that high-frequency information is critical for detecting subtle and short-term anomalies. Without this module, the model becomes less sensitive to rapid or minor changes, resulting in reduced detection accuracy.

Next, we evaluate the impact of removing the reinforcement learning rewards mechanism, *i.e.,* TAD-NET(–R). While the drop in AUC is less pronounced than when removing high-frequency amplification, it still indicates that reinforcement learning is important for helping the model adapt to temporal changes. Without this adaptive feedback, the model's ability to track evolving patterns is diminished.

The lowest AUC values are observed when both high-frequency amplification and reinforcement learning are removed, *i.e.,* TAD-NET(–B), highlighting the necessity of both components. The absence of high-frequency features limits the detection of transient anomalies, and the lack of reinforcement learning reduces adaptability, leading to the greatest performance degradation.

In summary, these ablation results show that both high-frequency amplification and reinforcement learning are essential for robust dynamic anomaly detection. Each component addresses a different aspect of the problem—capturing fine-grained changes and adapting to non-stationary environments—and their combination is crucial for achieving high AUC in dynamic graph scenarios.

## 5.4 PARAMETER SENSITIVITY STUDY

We analyze the sensitivity of TADNet to the hyperparameter $\alpha$, which controls the contribution of wavelet-based high-frequency features in node embeddings (Equation 9). We report AUC as $\alpha$ varies from 0 to 1 (Figure 4).

Results show that datasets respond differently: on WIKIPEDIA, larger $\alpha$ steadily improves AUC, indicating the benefit of emphasizing wavelet components. On REDDIT, performance rises quickly at small $\alpha$ then stabilizes, suggesting that moderate weighting is most effective. On MOOC, AUC remains flat, implying robustness to this parameter. Overall, TADNet is stable on most datasets,

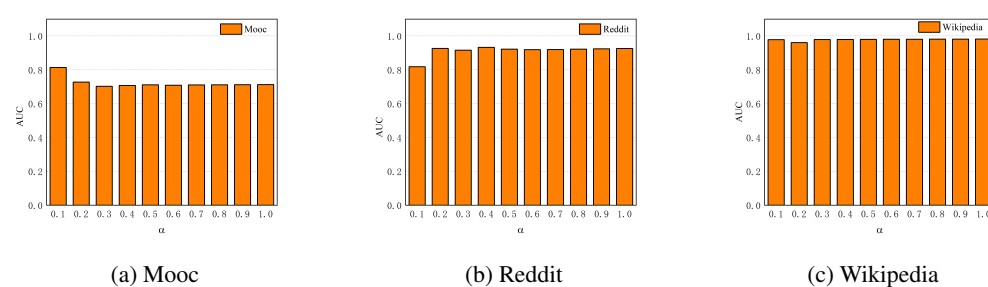

(a) Mooc          (b) Reddit          (c) Wikipedia

Figure 4: Comparison of AUC values for different $\alpha$ across datasets

with only REDDIT requiring mild tuning at low $\alpha$. Additional sensitivity studies and theoretical analysis are provided in Appendix J.

## 5.5 CONCEPT DRIFT RESISTANCE ANALYSIS

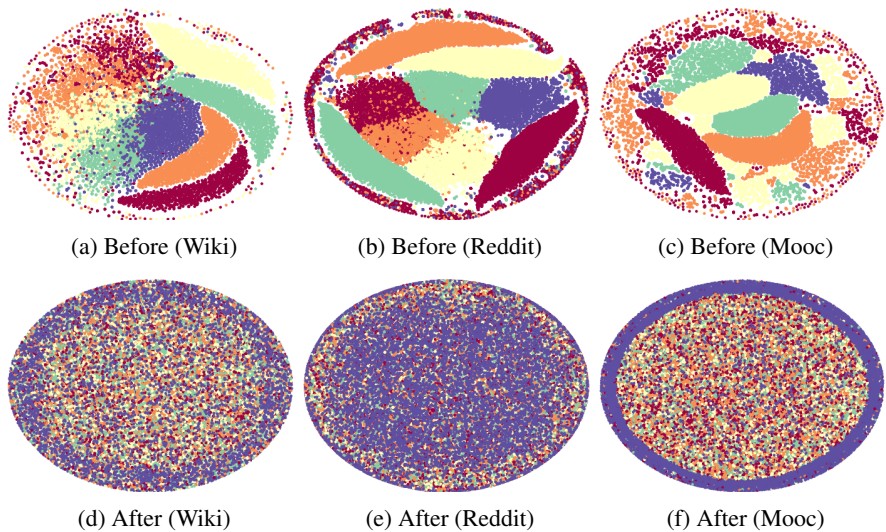

(a) Before (Wiki)     (b) Before (Reddit)     (c) Before (Mooc)

(d) After (Wiki)     (e) After (Reddit)     (f) After (Mooc)

Figure 5: Node feature distributions using t-SNE clustering. Different colors correspond to different snapshots.

To examine TAD-NET's robustness against concept drift in dynamic graphs, we visualize node embeddings across multiple time snapshots using t-SNE. Different colors denote temporal snapshots. **Before Training.** Figures 5a, 5b, and 5c show that embeddings from different time steps are well separated, indicating the model initially lacks temporal invariance and is vulnerable to distribution shifts. **After Training.** With temporal feature extraction, reinforced anomaly generation, and wavelet-based fusion, the embeddings (Figures 5d, 5e, 5f) become much more intermixed across time. This demonstrates that TAD-NET aligns node representations over time, capturing temporally robust features and mitigating drift effects. Overall, temporal extraction enhances stability, anomaly generation improves adaptability, and wavelet fusion preserves anomaly-relevant signals—together enabling effective resistance to concept drift.

## 6 CONCLUSION

We addressed anomaly detection in dynamic graphs under concept drift by proposing TAD-NET, a framework that combines temporal feature extraction, reinforced anomaly generation, and wavelet-based feature fusion. These modules enable TAD-NET to adapt to evolving networks and reliably detect anomalies, even with limited labeled data. Experiments on real-world datasets show that TAD-NET outperforms existing methods and remains robust as network conditions change.

## ETHICS STATEMENT

Our research focuses on anomaly detection in dynamic graph networks, such as social or communication networks, with the goal of identifying abnormal behaviors (e.g., fraud, fake accounts). We only use publicly available datasets and synthetic data for experiments, ensuring no personally identifiable information is exposed. The methods developed are intended to improve security and reliability of networked systems. We acknowledge that misuse of anomaly detection techniques may raise privacy or fairness concerns, and we encourage responsible application and adherence to relevant laws and regulations.

## REPRODUCIBILITY STATEMENT

We have taken steps to ensure the reproducibility of our results. All datasets used in our experiments are publicly available, and the main text provides detailed descriptions of data preprocessing, model architectures, hyperparameters, and training procedures. The code for our experiments is publicly available at `https://anonymous.4open.science/r/TAD-Net-B26A`.

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

## A    APPENDIX: USE OF LARGE LANGUAGE MODELS (LLMS)

ChatGPT (GPT-4) was utilized only as an auxiliary tool for improving English grammar, enhancing readability of the text, and debugging minor coding issues. It played no role in the design of the research, development of algorithms, or interpretation of results, which were fully conducted by the authors.

## B    BACKGROUND: PRELIMINARY CONCEPTS

### B.1    CONCEPT DRIFT IN DYNAMIC GRAPHS

Dynamic graphs, such as social networks, often experience evolving node attributes and structural patterns over time—a phenomenon known as *concept drift*. This presents a major challenge for anomaly detection, as models trained on historical data may become less effective when the underlying data distribution shifts.

In social networks, concept drift can arise from:

- **Behavioral Change:** Users change their interaction patterns (e.g., shifting from text to video posts).
- **Community Evolution:** Groups merge, split, or change membership, altering group characteristics.
- **Emergence of New Topics:** Trending topics or events cause sudden changes in user activity.
- **Platform Changes:** New features or policies shift user behavior (e.g., introduction of short videos).
- **Account Hacking:** Compromised accounts exhibit abrupt, atypical behavior.

These changes can degrade model performance, increasing false positives or negatives, and complicate model maintenance due to the need for frequent retraining and adaptation. Ignoring concept drift risks misidentifying normal or abnormal behaviors, undermining detection reliability.

### B.2    ADDRESSING CONCEPT DRIFT

To address concept drift in dynamic graphs, we combine two strategies: reinforced adversarial anomaly generation and high-frequency feature processing via discrete wavelet transform (DWT).

**Reinforced Adversarial Anomaly Generation**

Anomalies in dynamic graphs are rare and diverse, making them hard to model. Standard GANs often suffer from mode collapse, generating insufficiently varied anomalies. By integrating reinforcement learning into the GAN framework, the generator is incentivized to explore a broader range of outputs, improving diversity and realism. The generator receives feedback from the discriminator, guiding it to produce more representative anomalies.

Key benefits of reinforced GANs for anomaly generation:

- **Improved Exploration:** Reinforcement learning encourages discovery of diverse anomaly patterns.
- **Adaptive Generation:** The generator adapts to evolving data distributions.
- **Greater Diversity:** A wider variety of synthetic anomalies enhances detection of different anomaly types.

This approach helps overcome the scarcity of labeled anomalies by generating synthetic samples that augment the training set, improving model robustness.

**Discrete Wavelet Transform for Feature Fusion**    Concept drift can be gradual, abrupt, recurring, or incremental. To capture these, we use the *Discrete Wavelet Transform (DWT)*, which decomposes time-series data into:

- **Low-Frequency Components:** Long-term, stable trends.
- **High-Frequency Components:** Short-term, abrupt changes or anomalies.

DWT is effective for concept drift because it separates slow, gradual shifts (low-frequency) from sudden changes (high-frequency), allowing the model to:

- Detect rapid changes without interference from long-term trends.
- Preserve stable patterns while remaining sensitive to new or rare anomalies.

**Feature Fusion:** We combine high-frequency features from DWT with original features, enabling:

- **Adaptive Learning:** Robustness to sudden changes while retaining long-term knowledge.
- **Enhanced Sensitivity:** Improved detection of abrupt behavioral changes.
- **Long-Term Stability:** Retention of persistent patterns.

In summary, our approach integrates reinforced adversarial anomaly generation and DWT-based feature fusion to effectively address concept drift, enabling reliable detection of both gradual and sudden changes in dynamic graphs.

## C    NOTATIONS

Table 3 lists the notations used in this paper.

## D    TAD-NET WORKFLOW

Algorithm 1 summarizes the TAD-NET workflow. At each time step $t$, the model receives a dynamic graph snapshot $(X_t, A_t)$, where $X_t$ is the node feature matrix and $A_t$ the adjacency matrix. The Temporal Feature Extraction (TFE) module encodes temporal dynamics by computing node representations $H_t$ and their temporal differences $\Delta H_t$, highlighting abrupt behavioral changes. The Reinforced Anomaly Generation (RAG) module augments the training set with synthetic anomalies, improving detection of diverse patterns. The Wavelet-Enhanced Fusion Predictor (WFP) applies DWT to both real and synthetic features, fuses original and high-frequency components, and uses a neural network classifier to assign anomaly scores. The model is trained end-to-end on both real and generated data. During inference, the trained WFP outputs anomaly scores for each node, enabling robust anomaly detection as the graph evolves.

## E    TIME COMPLEXITY ANALYSIS

We analyze the time complexity of each module in the Temporal Anomaly Detection Network (TADNet): Temporal Feature Extractor (TFE), Reinforced Anomaly Generator (RAG), and Wavelet-Enhanced Fusion Predictor (WFP).

Table 3: Notations

| Symbol | Description |
|--------|-------------|
| $L_D$ | Discriminator loss function |
| $L_G$ | Generator loss function |
| $\gamma$ | Discount factor in RL |
| $L_{\text{CE}}$ | Cross-entropy loss |
| $\eta_d$ | Learning rate of the discriminator |
| $\eta_g$ | Learning rate of the generator |
| $N_{\text{gen}}$ | The number of generated anomalous samples |
| $N_t$ | The number of nodes at time step $t$ |
| $E_t$ | Edges in the graph at time step $t$ |

---

**Algorithm 1** TADNet Training Procedure

---

**Require:** Dynamic graph sequence $\{G_t\}_{t=1}^T$, node features $\{X_t\}_{t=1}^T$, learning rates $\eta_d, \eta_g, \eta$, iterations $I$
**Ensure:** Trained model parameters $W, \theta_g, \theta_d, \theta_f$
 1: **Phase 1: Temporal Feature Extraction**
 2: **for** $t = 1$ **to** $T$ **do**
 3:     Compute $H_t = \sigma(\tilde{D}^{-1/2}\tilde{A}\tilde{D}^{-1/2}X_tW)$ {Corresponds to Eq. 3}
 4:     **if** $t = 1$ **then**
 5:         Initialize $\Delta H_1 \leftarrow \mathbf{0}$ {No temporal difference for first snapshot}
 6:     **else**
 7:         Calculate $\Delta H_t$ using padding/truncation {Implements Eq. 4}
 8:     **end if**
 9:     Update $W$ via gradient descent on $H_t$
10: **end for**
11: **Phase 2: Reinforced Anomaly Generation**
12: **for** $epoch = 1$ **to** $I$ **do**
13:     Sample minibatch of real anomalies $X_a \subseteq \{\Delta H_t\}_{t=1}^T$
14:     Generate synthetic anomalies $\hat{X}_a = G(Z)$ where $Z \sim \mathcal{N}(0, I)$
15:     Compute discriminator loss $L_D$ via Eq. 5
16:     Update $\theta_d \leftarrow \theta_d - \eta_d \nabla_{\theta_d} L_D$
17:     Compute generator loss $L_G$ via Eq. 6
18:     Update $\theta_g \leftarrow \theta_g - \eta_g \nabla_{\theta_g} L_G$
19: **end for**
20: **Phase 3: Wavelet-Enhanced Prediction**
21: **for** $epoch = 1$ **to** $I$ **do**
22:     For each $v \in \{\Delta H_t\}_{t=1}^T \cup \hat{X}_a$:
23:         Apply DWT: $\mathtt{C} \leftarrow \mathrm{DWT}(v)$
24:         Extract $H_{\mathrm{high}}^v \leftarrow \mathtt{C}[1]$
25:         Compute $H_{\mathrm{fusion}}^v = v + \alpha H_{\mathrm{high}}^v$ {Implements Eq. 9}
26:     Compute $L_{\mathrm{CE}}$ over fused features {Using Eq. 10}
27:     Update $\theta_f$ via gradient descent on $L_{\mathrm{CE}}$
28: **end for**

---

### E.1 TEMPORAL FEATURE EXTRACTOR

The TFE processes a sequence of graph snapshots. Main operations:

- **GCN Layer:** For each time step, processing $X_t$ and $G_t$ takes $O(|E_t| + N_tFd)$, where $|E_t|$ is the number of edges, $N_t$ nodes, $F$ feature dimension, $d$ embedding dimension.
- **Feature Update:** Updating $X_t$ with $H_{t-1}$: $O(N_{t-1}d)$.
- **Relative Change:** Computing $H_t - H_{t-1}$: $O(N_t d)$.

For $T$ time steps, total complexity:

$$O\left(\sum_{t=1}^T (|E_t| + N_tFd + N_{t-1}d)\right). \tag{11}$$

### E.2 REINFORCED ANOMALY GENERATOR

RAG combines GAN and reinforcement learning:

- **Generator Forward Pass:** $O(N_{\mathrm{gen}}d)$, where $N_{\mathrm{gen}}$ is the number of generated anomalies.
- **Discriminator Training:** $O(Md)$, $M$ is the number of real anomalies.
- **Backpropagation:** $O(N_{\mathrm{gen}}d + Md)$.

Total for $I$ iterations:

$$O\left(I(M + N_{\mathrm{gen}})d\right). \tag{12}$$

### E.3 WAVELET-ENHANCED FUSION PREDICTOR

WFP applies wavelet transforms and classification:

- **DWT:** $O((N_t + M)d \log d)$ for all feature vectors.
- **Feature Fusion:** $O((N_t + M)d)$.
- **Classification:** $O((N_t + M)dC)$, $C$ is the number of classifier layers.

Total complexity:

$$O((N_t + M)d \log d + (N_t + M)dC). \tag{13}$$

### E.4 OVERALL COMPLEXITY OF TAD-NET

Summing all modules, the total time complexity over $T$ time steps is:

$$O\Bigg( \sum_{t=1}^{T} \Big( |E_t| + N_t d(F + 1) + N_{t-1}d \Big) + I(M + N_{\text{gen}})d$$

$$+ (N_t + M)d \log d + (N_t + M)dC \Bigg). \tag{14}$$

Typically:

- TFE dominates for large graphs with many nodes/edges.
- RAG and WFP may dominate for smaller graphs or when generating many synthetic anomalies or applying large DWTs.

# F THEORETICAL ANALYSES

In this section, we provide a theoretical justification for the core components of TAD-NET.

## F.1 TEMPORAL FEATURE EXTRACTOR

This section provides a clear theoretical justification for the Temporal Feature Extractor (TFE) module of TAD-NET. We show that temporal differencing suppresses smooth (normal) variations in dynamic graphs while amplifying anomaly-induced deviations, thus enabling robust anomaly detection.

**Motivation.** When node representations in a dynamic graph evolve smoothly, the temporal difference of node embeddings, $\Delta H_t$, remains bounded. In contrast, anomalies cause abrupt changes in the input, which propagate through the encoder and result in significantly larger values of $\|\Delta H_t\|_F$. Thus, temporal differencing naturally highlights anomalous behavior.

Let $X_t \in \mathbb{R}^{N_t \times d}$ be the node feature matrix at time $t$, and $H_t = f(A, X_t) \in \mathbb{R}^{N_t \times h}$ the output of a graph convolution layer, where $A$ is the (possibly self-loop augmented) adjacency matrix at time $t$. If $N_t \neq N_{t-1}$, we compare $H_t$ and $H_{t-1}$ over the first $\min(N_t, N_{t-1})$ rows. For a full-dimension difference, we use a non-expansive padding operator $\mathtt{Pad}(\cdot, n)$ and define:

$$\Delta H_t := \begin{cases} H_t - H_{t-1}[1:N_t, :], & \text{if } N_t \leq N_{t-1}, \\ H_t - \mathtt{Pad}(H_{t-1}, N_t), & \text{if } N_t > N_{t-1}. \end{cases}$$

**Assumption 4.1** (Lipschitz temporal evolution and non-expansive padding). *For* normal *evolution, the feature sequence is $L_X$-Lipschitz in time: $\|X_t - X_{t-1}\|_F \leq L_X$. Let one GCN layer be $f(A, X) = \sigma\big(\tilde{D}^{-1/2} \tilde{A} \tilde{D}^{-1/2} XW\big)$, where $\sigma$ is $L_\sigma$-Lipschitz, $\|\tilde{D}^{-1/2} \tilde{A} \tilde{D}^{-1/2}\|_2 \leq L_A$, and $W$ is a trainable weight matrix. The padding operator is non-expansive: for any $U, V$ and any $n$, $\|\mathtt{Pad}(U, n) - \mathtt{Pad}(V, n)\|_F \leq \|U - V\|_F$.*

**Assumption F.2** (Optional local gain (for lower bounds)). *There exists a* local *constant $\mu_f > 0$ (possibly data-dependent) such that for inputs on the line segment between $X_{t-1}$ and $X_t$,*

$$\|f(A, U) - f(A, V)\|_F \geq \mu_f \|U - V\|_F, \qquad U, V \in \{X_{t-1} + s(X_t - X_{t-1}) : s \in [0, 1]\}.$$

*This holds, for example, when $\sigma$ is piecewise linear (e.g., ReLU) and the segment stays in one linear region so that $f$ reduces to a linear map with smallest singular value at least $\mu_f$.*

**Lemma 4.1** (Stability of one-step embedding). *Under Assumption 4.1, the mapping $X \mapsto H = f(A, X)$ is $L_f$-Lipschitz in Frobenius norm with $L_f \leq L_\sigma L_A \|W\|_2$. That is, for any $X, X'$,*

$$\|f(A, X) - f(A, X')\|_F \leq L_f \|X - X'\|_F.$$

*Proof.* By submultiplicativity and Lipschitzness of $\sigma$,

$$\|f(A, X) - f(A, X')\|_F = \|\sigma(\tilde{S}XW) - \sigma(\tilde{S}X'W)\|_F \leq L_\sigma \|\tilde{S}(X - X')W\|_F \leq L_\sigma \|\tilde{S}\|_2 \|W\|_2 \|X - X'\|_F,$$

where $\tilde{S} = \tilde{D}^{-1/2} \tilde{A} \tilde{D}^{-1/2}$ and $\|\tilde{S}\|_2 \leq L_A$. Set $L_f := L_\sigma L_A \|W\|_2$. $\qquad\square$

**Proposition F.1** (Bounded temporal difference for normal evolution). *Under Assumption 4.1, the temporal difference of embeddings satisfies*

$$\|\Delta H_t\|_F \leq L_f L_X + R_t,$$

*where $R_t = 0$ if $N_t \leq N_{t-1}$, and otherwise*

$$R_t \leq L_f \|\mathtt{Pad}(X_{t-1}, N_t) - X_{t-1}\|_F.$$

*Proof.* If $N_t \leq N_{t-1}$, then for the first $N_t$ rows,

$$\|\Delta H_t\|_F = \|f(A, X_t) - f(A, X_{t-1})\|_F \leq L_f \|X_t - X_{t-1}\|_F \leq L_f L_X.$$

If $N_t > N_{t-1}$, add and subtract $f(A, \texttt{Pad}(X_{t-1}, N_t))$ and apply Lemma 4.1 and the non-expansiveness of padding:

$$\|\Delta H_t\|_F = \|f(A, X_t) - f(A, \texttt{Pad}(X_{t-1}, N_t)) + f(A, \texttt{Pad}(X_{t-1}, N_t)) - f(A, \texttt{Pad}(H_{t-1}, N_t))\|_F$$

$$\leq L_f \|X_t - \texttt{Pad}(X_{t-1}, N_t)\|_F + \underbrace{\|f(A, \texttt{Pad}(X_{t-1}, N_t)) - \texttt{Pad}(H_{t-1}, N_t)\|_F}_{=0}$$

$$\leq L_f \left(\|X_t - X_{t-1}\|_F + \|\texttt{Pad}(X_{t-1}, N_t) - X_{t-1}\|_F\right) \leq L_f L_X + R_t.$$

$\square$

**Theorem 4.1** (Detection margin under anomaly perturbation). *Suppose an anomaly increases the input temporal jump by at least $\delta > 0$, i.e., $\|X_t - X_{t-1}\|_F \geq L_X + \delta$. If, moreover, the encoder satisfies the local gain condition in Assumption F.2, then $\|\Delta H_t\|_F \geq \mu_f (L_X + \delta) - R_t$. Therefore the excess over the normal bound $\tau_t = L_f L_X + R_t$ obeys $\|\Delta H_t\|_F - \tau_t \geq \mu_f \delta - (L_f - \mu_f) L_X - 2R_t$. In particular, a sufficient condition for a positive detection margin is: $\mu_f \delta > (L_f - \mu_f) L_X + 2R_t$.*

*Proof.* If $N_t \leq N_{t-1}$, Assumption F.2 on the segment $[X_{t-1}, X_t]$ implies

$$\|\Delta H_t\|_F = \|f(A, X_t) - f(A, X_{t-1})\|_F \geq \mu_f \|X_t - X_{t-1}\|_F \geq \mu_f (L_X + \delta).$$

If $N_t > N_{t-1}$, insert and subtract $f(A, \texttt{Pad}(X_{t-1}, N_t))$ and use triangle inequality, local gain on the segment $[\texttt{Pad}(X_{t-1}, N_t), X_t]$, and non-expansiveness of padding to get

$$\|\Delta H_t\|_F \geq \mu_f \|X_t - \texttt{Pad}(X_{t-1}, N_t)\|_F - R_t \geq \mu_f \left(\|X_t - X_{t-1}\|_F - \|\texttt{Pad}(X_{t-1}, N_t) - X_{t-1}\|_F\right) - R_t,$$

which yields the stated inequality since $\|X_t - X_{t-1}\|_F \geq L_X + \delta$. Subtracting $\tau_t = L_f L_X + R_t$ and rearranging gives the claimed margin bound. $\square$

*Remark* F.1 (What each assumption provides). Assumption 4.1 is sufficient to obtain an *upper* bound for normal evolution and thus a sound threshold test. Any *lower* bound (i.e., a guaranteed margin under anomalies) requires additional structure such as Assumption F.2 (a local bi-Lipschitz property).

*Remark* F.2 (Multilayer encoders). For $L$ stacked layers with Lipschitz constants $L_{f,\ell}$, the composite map is $L_f^{(\text{stack})} \leq \prod_{\ell=1}^{L} L_{f,\ell}$. The proofs carry over verbatim by replacing $L_f$ with $L_f^{(\text{stack})}$ and adjusting $\mu_f$ accordingly (e.g., the smallest local gain along the stack).

*Remark* F.3 (Time-varying graphs). If $A_t$ varies with $t$ but the operator norm of the normalized adjacency is uniformly bounded $\|\tilde{D}_t^{-1/2} \tilde{A}_t \tilde{D}_t^{-1/2}\|_2 \leq L_A$, the stability lemma and proposition remain valid with the same $L_f$; an extra term involving $\|\tilde{S}_t - \tilde{S}_{t-1}\|_2$ can be included if one wishes to account explicitly for graph dynamics.

*Remark* F.4 (On padding term $R_t$). If $N_t \leq N_{t-1}$, then $R_t = 0$. If $N_t > N_{t-1}$, $R_t$ scales with how many new rows are padded and the magnitude of padding values. Using zero-padding or duplicated last-observation padding preserves non-expansiveness.

*Remark* F.5 (Practical thresholds). In practice, one can estimate $L_f L_X + R_t$ from a calibration window via high quantiles (e.g., 95%) of $\|\Delta H_t\|_F$ during known-normal periods, and then flag $\|\Delta H_t\|_F$ above that empirical threshold.

### F.2 THEORETICAL DETAILS OF REINFORCED ANOMALY GENERATOR

This appendix provides detailed theoretical justification for the Reinforced Anomaly Generator (RAG) introduced in Section 4.2, including policy-gradient updates, entropy-regularized objectives, and guarantees on anomaly coverage.

#### F.2.1 NOTATION AND SETUP

- $\mathcal{S} \subset \mathbb{R}^{d'}$: anomaly feature state space extracted from the Temporal Feature Extractor.
- $P_{\text{anom}}$: true (unknown) anomaly distribution.
- $P_{\text{data}}$: empirical distribution of observed anomalies used to train the generator.

- Generator $G_\theta$ defines a parameterized policy $\pi_\theta(a|s)$ mapping a state $s \in \mathcal{S}$ to a generated anomaly $a \in \mathcal{S}$.

- Discriminator $D_\phi : \mathcal{S} \to [0,1]$ outputs the probability of a sample being real and provides a reward $r(a) = D_\phi(a)$.

- $\hat{P}_\theta$: induced distribution of generated anomalies under policy $\pi_\theta$.

### F.2.2 Unbiased Policy-Gradient

**Lemma F.2** (Unbiased policy-gradient, rigorous version). *Let $a = G_\theta(Z)$ with $Z \sim P_z$, and reward $r(a) = \log D_\phi(a)$. Then*

$$\nabla_\theta \mathbb{E}_{a \sim \pi_\theta}[r(a)] = \mathbb{E}_{a \sim \pi_\theta}[\nabla_\theta \log \pi_\theta(a)\, r(a)].$$

*Proof.* Assume $r(a)$ does not depend on $\theta$, and $\pi_\theta(a)$ is differentiable with respect to $\theta$ with sufficient integrability to allow exchanging gradient and integral. By definition of expectation:

$$\mathbb{E}_{a \sim \pi_\theta}[r(a)] = \int_{\mathcal{A}} r(a)\pi_\theta(a)\, da.$$

Taking the gradient and exchanging it with the integral gives

$$\nabla_\theta \mathbb{E}_{a \sim \pi_\theta}[r(a)] = \int_{\mathcal{A}} r(a)\, \nabla_\theta \pi_\theta(a)\, da.$$

Using the identity $\nabla_\theta \pi_\theta(a) = \pi_\theta(a)\nabla_\theta \log \pi_\theta(a)$, we obtain

$$\int_{\mathcal{A}} r(a)\, \nabla_\theta \pi_\theta(a)\, da = \int_{\mathcal{A}} r(a)\pi_\theta(a)\nabla_\theta \log \pi_\theta(a)\, da = \mathbb{E}_{a \sim \pi_\theta}[r(a)\nabla_\theta \log \pi_\theta(a)].$$

This establishes the lemma. $\qquad\square$

**Remark.** In our GAN-RL framework, the reward is defined as $r(a) = \log D_\phi(a)$. Although $D_\phi$ is trained concurrently with the generator, at each generator update step $D_\phi$ is treated as fixed. Therefore, within the gradient computation $\nabla_\theta \mathbb{E}_{a \sim \pi_\theta}[r(a)]$, the reward $r(a)$ is independent of $\theta$. This ensures that the standard policy-gradient derivation remains valid and yields an unbiased estimate of the gradient with respect to the generator parameters.

### F.2.3 Reward-Regularized Adversarial Objective

To encourage diversity in generated anomalies, we introduce an entropy-regularized objective for the generator. Let the generator's policy be $q_\theta(a)$ over generated samples $a$, and define a reward

$$r(a) = \log D_\phi(a) + \beta u(a),$$

where $D_\phi(a)$ is the discriminator output, $\beta > 0$ is a weighting factor, and $u(a) = -\log q_\theta(a)$ is an entropy-related term. The intuition is that maximizing $\mathbb{E}[u(a)]$ encourages the generator to produce a more diverse set of samples, avoiding mode collapse.

**Proposition F.2** (Entropy-regularized GAN objective). *Under the above definitions, the generator's reward objective can be equivalently written as*

$$\max_\theta \mathbb{E}_{a \sim q_\theta}[r(a)] = \max_\theta \mathbb{E}_{a \sim q_\theta}[\log D_\phi(a)] + \beta H(q_\theta),$$

*where $H(q_\theta) = -\mathbb{E}_{a \sim q_\theta}[\log q_\theta(a)]$ is the Shannon entropy of the generator distribution.*

*Proof.* By substituting $u(a) = -\log q_\theta(a)$ into the reward:

$$\mathbb{E}_{a \sim q_\theta}[r(a)] = \mathbb{E}_{a \sim q_\theta}[\log D_\phi(a)] + \beta \mathbb{E}_{a \sim q_\theta}[-\log q_\theta(a)] = \mathbb{E}_{a \sim q_\theta}[\log D_\phi(a)] + \beta H(q_\theta),$$

which establishes the equivalence. $\qquad\square$

**Interpretation.** The first term, $\mathbb{E}[\log D_\phi(a)]$, is the standard GAN objective that encourages generating realistic samples. The second term, $\beta H(q_\theta)$, explicitly rewards the generator for maintaining high entropy in its output distribution, which promotes diversity and prevents the generator from collapsing to a few high-probability modes. By tuning $\beta$, one can balance fidelity to the discriminator with diversity in the generated anomalies. This formulation integrates naturally into our reinforcement-learning-inspired generator update, providing both realism and coverage in anomaly synthesis.

### F.2.4 MINIMUM MASS GUARANTEE ON DATA MODES

The reinforced anomaly generator in our framework is expected to produce diverse synthetic anomalies, rather than collapsing onto only a few specific patterns. In dynamic graphs, such diversity corresponds to covering multiple *data modes*, each reflecting a distinct type of anomalous feature deviation. To theoretically ensure that the generator does not neglect any mode, we provide a *minimum mass guarantee* under the entropy-regularized adversarial objective.

**Theorem F.2** (Minimum mass guarantee). *Assume $P_{\text{data}}$ decomposes into disjoint measurable regions $\{\mathcal{M}_k\}_{k=1}^{K}$ with $P_{\text{data}}(\mathcal{M}_k) \geq \delta_k > 0$. Let $\beta > 0$ and assume the discriminator $D_\phi$ is strictly positive and bounded on each $\mathcal{M}_k$: $m_k \leq D_\phi(a) \leq M_k$ for all $a \in \mathcal{M}_k$. Then any stationary point $q_\theta^*$ of the entropy-regularized objective satisfies*

$$q_\theta^*(\mathcal{M}_k) \geq \frac{\beta}{\beta + \log M_k - \log m_k}\, \delta_k.$$

*Proof.* We analyze the entropy-regularized objective by partitioning $q_\theta$ across data modes. Define the mode probability and conditional distribution as

$$q_k := q_\theta(\mathcal{M}_k), \quad q_\theta(a|\mathcal{M}_k) := \frac{q_\theta(a)}{q_k}.$$

This allows us to decompose the objective into three interpretable terms:

$$J(q_\theta) = \sum_{k=1}^{K} q_k \mathbb{E}_{a \sim q_\theta(\cdot|\mathcal{M}_k)}[\log D_\phi(a)] + \beta \sum_{k=1}^{K} q_k H(q_\theta(\cdot|\mathcal{M}_k)) - \beta \sum_{k=1}^{K} q_k \log q_k,$$

where the first term reflects discriminator alignment, the second captures within-mode entropy, and the third penalizes extremely small mode probabilities.

Now consider the stationary condition w.r.t. each $q_k$. Differentiating $J$ with respect to $q_k$ yields

$$\frac{\partial J}{\partial q_k} = \mathbb{E}_{a \sim q_\theta(\cdot|\mathcal{M}_k)}[\log D_\phi(a)] - \beta(1 + \log q_k) = 0,$$

which gives the closed-form stationary point

$$q_k^* = \exp\left( \frac{\mathbb{E}_{a \sim q_\theta(\cdot|\mathcal{M}_k)}[\log D_\phi(a)]}{\beta} - 1 \right).$$

Since the discriminator is bounded on each $\mathcal{M}_k$, we use

$$\log m_k \leq \mathbb{E}_{a \sim q_\theta(\cdot|\mathcal{M}_k)}[\log D_\phi(a)] \leq \log M_k$$

to obtain the lower bound

$$q_k^* \geq \exp\left( \frac{\log m_k}{\beta} - 1 \right).$$

Finally, incorporating the data measure $\delta_k$ of each mode, we derive the guaranteed minimum allocation:

$$q_\theta^*(\mathcal{M}_k) \geq \frac{\beta}{\beta + \log M_k - \log m_k}\, \delta_k,$$

ensuring that no data mode is neglected during training. $\square$

### F.2.5 GENERALIZATION UNDER TEMPORAL EVOLUTION

**Lemma F.3** (GAN mode collapse). *A standard GAN trained solely on $P_{\text{data}}$ produces samples concentrated in high-density regions, failing to cover rare or unseen anomalies. In particular, if $\hat{P}_\theta$ is the generator distribution, then*

$$\mathrm{supp}(\hat{P}_\theta) \subseteq \mathrm{supp}(P_{data}),$$

*leading to mode collapse and poor anomaly coverage.*

**Theorem F.3** (RAG generalization). *Assume the generator policy is Gaussian,*

$$\pi_\theta(a|s) = \mathcal{N}(\mu_\theta(s), \Sigma_\theta(s)),$$

*so that $\pi_\theta(a|s) > 0$ for all $a \in \mathbb{R}^d$. Let $r(a) = D_\phi(a)$ be the reward, and suppose there exists a region $\Omega$ such that*

$$P_{data}(\Omega) = 0, \quad but \quad P_{anom}(\Omega) > 0, \quad and \quad r(a) \geq c > 0 \ \ \forall a \in \Omega.$$

*Then under policy gradient updates*

$$\nabla_\theta J(\pi_\theta) = \mathbb{E}_{s,a\sim\pi_\theta}\big[\nabla_\theta \log \pi_\theta(a|s)\, r(a)\big],$$

*the generator distribution $\hat{P}_\theta$ assigns strictly positive probability mass to $\Omega$ after finitely many updates. Hence, $\mathrm{supp}(\hat{P}_\theta)$ expands beyond $\mathrm{supp}(P_{data})$, improving temporal generalization to unseen anomalies.*

*Proof.* Since $\pi_\theta(a|s)$ is Gaussian with non-degenerate covariance $\Sigma_\theta(s)$, its support is the entire $\mathbb{R}^d$. Thus, for any measurable $\Omega \subseteq \mathbb{R}^d$,

$$\pi_\theta(a \in \Omega \mid s) > 0.$$

By assumption, $r(a) \geq c > 0$ for all $a \in \Omega$. Therefore the policy gradient satisfies

$$\nabla_\theta J(\pi_\theta) = \mathbb{E}_{s,a\sim\pi_\theta}\big[\nabla_\theta \log \pi_\theta(a|s)r(a)\big] \ \geq \ c\,\mathbb{E}_{s,a\sim\pi_\theta}\big[\nabla_\theta \log \pi_\theta(a|s)\,\mathbf{1}_\Omega(a)\big].$$

The expectation is nonzero since $\pi_\theta(a|s)$ has positive density in $\Omega$. Thus, gradient ascent increases $\pi_\theta(a|s)$ for $a \in \Omega$. Equivalently, the induced generator distribution $\hat{P}_\theta$ places increasing probability mass on $\Omega$:

$$\hat{P}_\theta^{t+1}(\Omega) > \hat{P}_\theta^t(\Omega), \quad \forall t,$$

until convergence. Consequently, after finitely many updates, $\hat{P}_\theta(\Omega) > 0$ even though $P_{\mathrm{data}}(\Omega) = 0$.

Since the reward $r(a)$ is updated over time to reflect evolving anomalies, this expansion property holds at each timestep, allowing $\hat{P}_\theta$ to adapt to temporally shifting anomaly distributions. $\square$

**Remark.** This theorem formalizes that combining adversarial learning with RL-guided exploration guarantees support expansion beyond the training data distribution, thereby mitigating GAN mode collapse and enhancing robustness to temporal evolution.

F.3    THEORETICAL DETAILS OF WAVELET-ENHANCED FUSION PREDICTOR

This appendix formalizes the theoretical properties of the WFP module introduced in Section 4.3.

**Assumption F.3** (Signal model and wavelet separation). *Each feature vector $v$ admits a decomposition*

$$v = s + a,$$

*where $s$ lies in the low-pass subspace and $a$ in the high-pass subspace of an orthonormal DWT basis.*

**Lemma F.4** (Energy preservation and separation). *Under Assumption F.3, Parseval's theorem implies*

$$\|v\|_2^2 = \|s\|_2^2 + \|a\|_2^2 \quad and \quad H_{high}^v = a.$$

**Proposition F.3** (SNR amplification). *Let $H_{\mathrm{fusion}}^v = v + \alpha a$ with $\alpha \geq 0$. Then*

$$\frac{\|Proj_{\mathrm{high}}(H_{\mathrm{fusion}}^v)\|_2}{\|Proj_{\mathrm{low}}(H_{\mathrm{fusion}}^v)\|_2} = \frac{(1+\alpha)\|a\|_2}{\|s\|_2} \geq \frac{\|a\|_2}{\|s\|_2}.$$

**Theorem F.4** (Improved detection under linear scoring). *Consider a linear detector $g(v) = w^\top a$, $\|w\|_2 = 1$. Under Assumption F.3, replacing $v$ by $H_{\mathrm{fusion}}^v$ scales the anomaly score to $g_\alpha(v) = (1+\alpha)g(v)$, while the low-pass contribution is unchanged. Therefore, for any fixed false-positive rate, the true-positive rate is non-decreasing in $\alpha$.*

*Proof.* The high-frequency projection scales linearly with $(1+\alpha)$. For any anomaly with $g(v) \neq 0$, this monotone scaling preserves or improves separability between normal and anomalous samples under linear scoring. $\square$

## G   ADDITIONAL EXPERIMENTAL SETTINGS

In this section, we provide additional details about the experimental settings.

### G.1   EXPERIMENTAL SETUP

The value of $\alpha$ is varied from 0 to 1 in increments of 0.1, and AUC is measured on three datasets: Wikipedia, Reddit, and Mooc. Each experiment is conducted three times to mitigate randomness, and the average AUC is reported. The experiments were conducted using NVIDIA GeForce RTX 4090 (24GB GDDR5X)-GPU. All experiments were run on a single GPU setup with batch sizes optimized for the 10GB memory capacity. The implementation leverages PyTorch's GPU acceleration for both the GCN operations and GAN training components.

### G.2   DATASETS

We use following datasets for the experiments:

**(i) Wikipedia Wang et al. (2020):** This dataset tracks user edits on wiki pages. Anomalous nodes represent users who suddenly increase their editing frequency or switch topics, leading to shifts in editing patterns. Concept drift occurs as users' behavior changes over time, making anomaly detection more challenging.

**(ii) Reddit Nguyen et al. (2020):** This dataset records user interactions in subreddits, including posts, comments, and voting activities. Anomalous users are those whose posting behavior drastically changes, such as posting too frequently or shifting focus to new topics. Concept drift occurs as trends or topics within subreddits evolve.

**(iii) Mooc Toghani et al. (2022):** This dataset logs student interactions on MOOC platforms. Anomalous students exhibit unusual engagement patterns, such as increased activity during exams or reduced participation during breaks. Concept drift arises from changes in student behavior and course content over time.

### G.3   BASELINES

In this section, we introduce the baselines used in the experiments.

**(i) TGAT Xu et al. (2020a)** utilizes the self-attention mechanism and introduces an innovative time-encoding technique based on Bochner's theorem from harmonic analysis.

**(ii) GDN Ding et al. (2021b)** employs a limited number of labeled anomalies to ensure statistically significant distinctions between abnormal and normal nodes.

**(iii) SAD Tian et al. (2023)** is a comprehensive anomaly detection framework tailored for dynamic graphs. It integrates a time-equipped memory bank with a pseudo-label contrastive learning module, effectively harnessing large unlabeled samples to identify anomalies within graph streams.

**(iv) TADDY Liu et al. (2021)** formulates a node encoding that encapsulates both spatial and temporal knowledge. It utilizes a solitary transformer model to grasp the interlinked spatial-temporal information.

**(v) MAMF Hong et al. (2025)** leverages Generative Adversarial Models (GANs) to augment the training with synthetic anomaly samples for learning anomaly patterns and combines meta-learning to combat concept drift.

These methods serve as baselines for comparison with our proposed framework to assess its effectiveness and performance in anomaly detection. By evaluating our framework against these diverse techniques, we can demonstrate its superiority and contribution.

### G.4   EVALUATION METRICS

We use the following metrics for the experiments:

Using AUC as the primary metric in the ablation study allows a consistent comparison with baseline tasks. However, our extended evaluation using F1-score, AUPR, and Precision adds a richer perspective on TADNet's performance in different aspects. Specifically:

**Precision**: Shows the model's ability to correctly identify anomalies among the predicted anomalies. High precision indicates a lower rate of false positives, suggesting that TADNet is adept at distinguishing true anomalies from normal data under ablation settings.

The precision metric is mathematically defined as:

$$\text{Precision} = \frac{\text{TP}}{\text{TP} + \text{FP}} \tag{15}$$

where TP (True Positives) is the number of correctly identified anomalies, and FP (False Positives) is the number of normal instances incorrectly classified as anomalies. High precision indicates that the model produces few false alarms when predicting anomalies.

**F1-Score**: The F1-score is the harmonic mean of precision and recall, offering a single metric that balances the trade-off between detecting true anomalies and avoiding false alarms. It is especially useful for evaluating performance on imbalanced datasets, where both false positives and false negatives are important. The F1-score is calculated as:

$$\text{F1} = 2 \cdot \frac{\text{Precision} \times \text{Recall}}{\text{Precision} + \text{Recall}} \tag{16}$$

where recall is defined as:

$$\text{Recall} = \frac{\text{TP}}{\text{TP} + \text{FN}} \tag{17}$$

Here, TP (True Positives) denotes correctly detected anomalies, and FN (False Negatives) denotes missed anomalies. By summarizing both precision and recall, the F1-score provides a comprehensive assessment of the model's anomaly detection capability across different ablation settings.

**AUPR**: The Area Under the Precision-Recall Curve (AUPR) is especially important for evaluating models on imbalanced datasets, where anomalies are rare. A high AUPR indicates that the model maintains strong detection performance even when the feature set or reward mechanisms are reduced, demonstrating robustness in sparse anomaly scenarios.

AUPR is calculated by plotting precision versus recall at different threshold values and measuring the area under this curve. For a set of thresholds $\{t_i\}$, AUPR can be estimated using the trapezoidal rule as follows:

$$\text{AUPR} = \sum_{i=1}^{n-1} (\text{Recall}_{i+1} - \text{Recall}_i) \cdot \frac{\text{Precision}_{i+1} + \text{Precision}_i}{2} \tag{18}$$

where $\text{Precision}_i$ and $\text{Recall}_i$ are the precision and recall at threshold $t_i$. This provides a single value summarizing the trade-off between precision and recall across all thresholds.

## H ADDITIONAL ABLATION EXPERIMENT

### H.1 WIKIPEDIA DATASET ANALYSIS

- **TAD-NET(–H)** shows significant drops in precision (Fig. 6a) and AUPR (Fig. 6c), demonstrating that high-frequency feature amplification is crucial for detecting subtle anomalies in dynamic graphs.
- **TAD-NET(–R)** exhibits reduced F1-score (Fig. 6b), confirming that the reinforcement learning mechanism is essential for generating diverse anomaly samples that improve model generalization.
- **TAD-NET(–B)** performs worst across all metrics, highlighting the complementary nature of these components - their combined removal causes the most severe performance degradation.

The consistent performance hierarchy TAD-NET ¿ (–H), (–R) ¿ (–B) reveals:

- High-frequency features (WFP module) are particularly effective for precision-oriented tasks

- Reinforcement-based generation (RAG module) significantly boosts recall and overall balanced performance

- The full model's synergy between these components provides optimal anomaly detection in dynamic environments

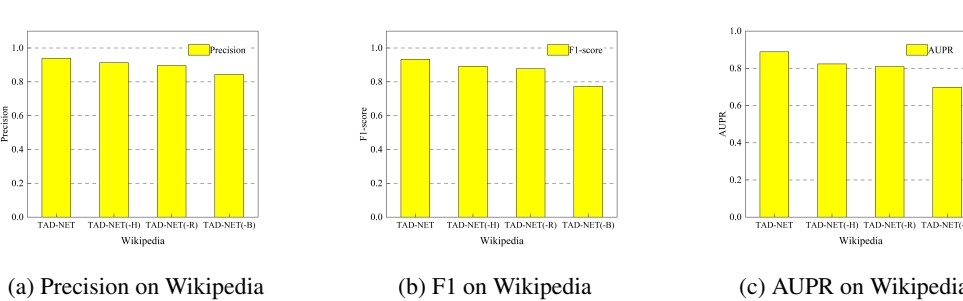

| (a) Precision on Wikipedia | (b) F1 on Wikipedia | (c) AUPR on Wikipedia |

Figure 6: Extended evaluations on Wikipedia dataset

## H.2    REDDIT DATASET ANALYSIS

- **TAD-NET(–H)** shows notable decreases in precision (Fig. 7a) and AUPR (Fig. 7c), proving that high-frequency feature extraction is vital for identifying nuanced anomalies in Reddit's rapidly evolving discussion threads.

- **TAD-NET(–R)** demonstrates declines in F1-score (Fig. 7b), verifying that the reinforcement learning component is critical for producing varied anomaly examples that enhance model adaptability to Reddit's diverse content patterns.

- **TAD-NET(–B)** exhibits the poorest performance across all metrics, emphasizing the interdependent relationship between these mechanisms - their simultaneous elimination leads to the most substantial performance deterioration.

The consistent performance ranking TAD-NET ¿ (–H), (–R) ¿ (–B) indicates:

- High-frequency analysis (WFP module) is especially valuable for precise anomaly detection in Reddit's volatile content environment

- Reinforcement-augmented sample generation (RAG module) substantially improves comprehensive detection capability

- The complete model's integrated approach delivers superior anomaly identification in Reddit's dynamic interaction networks

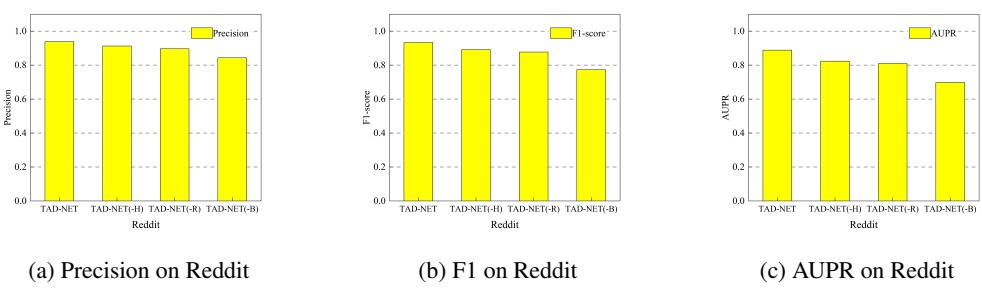

| (a) Precision on Reddit | (b) F1 on Reddit | (c) AUPR on Reddit |

Figure 7: Extended evaluations on Reddit dataset

### H.3 Mooc Dataset Analysis

- **TAD-Net(–H)** displays marked reductions in precision (Fig. 8a) and AUPR (Fig. 8c), confirming that high-frequency feature analysis is essential for detecting subtle anomalous patterns in Mooc's complex network structures.

- **TAD-Net(–R)** reveals decreased F1-score (Fig. 8b), establishing that the reinforcement learning framework is indispensable for creating comprehensive anomaly samples that strengthen model robustness on Mooc's diverse data.

- **TAD-Net(–B)** shows the weakest performance across all evaluation metrics, underscoring the synergistic relationship between these components - their joint removal results in the most significant performance decline.

The consistent performance gradient TAD-Net>(–H)>(–R)>(–B) demonstrates:

- High-frequency feature processing (WFP module) is particularly effective for precise anomaly identification in Mooc's specialized network environment

- Reinforcement-enhanced generation (RAG module) dramatically improves overall detection reliability

- The complete model's integrated architecture provides optimal anomaly recognition capabilities for Mooc's unique dataset characteristics

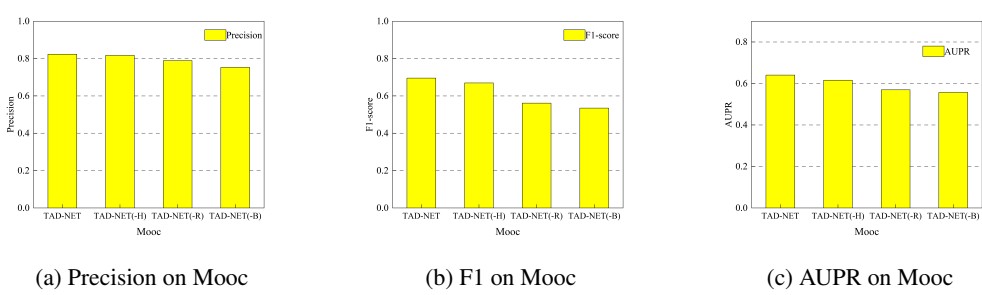

     (a) Precision on Mooc          (b) F1 on Mooc          (c) AUPR on Mooc

Figure 8: Extended evaluations on Mooc dataset.

## I Detailed Analysis of Parameter Sensitivity

In this appendix, we present a detailed analysis focusing on the hyperparameter $\alpha$ and its impact on AUC performance, as a supplementary exploration following the parameter analysis presented in the main text. While the main text primarily examines the relationship between $\alpha$ and AUC, this appendix delves deeper into the sensitivity of AUC to variations in $\alpha$ across different datasets. Additional metrics will be analyzed in subsequent sections to provide a more comprehensive evaluation.

### I.1 Theoretical Motivation

The hyperparameter $\alpha$ is designed to balance the contribution of wave features in the node embedding process. Intuitively, increasing $\alpha$ enhances the model's sensitivity to high-frequency features, which are crucial for detecting fine-grained anomalies. However, excessively large values may introduce noise, while overly small values may fail to capture rapid variations. Hence, identifying the optimal $\alpha$ is vital for achieving balanced and accurate anomaly detection.

### I.2 Results and Interpretation

**Wikipedia Dataset:** The AUC shows a steady increase as $\alpha$ rises, peaking at 0.9821 when $\alpha = 1.0$. This positive correlation indicates that the model benefits from incorporating more wave features, which effectively capture fine-grained temporal changes inherent in Wikipedia's dynamic graph structure.

**Reddit Dataset:** The AUC increases sharply at lower values of $\alpha$ (0.0 to 0.2), reaching a peak at 0.9261. This trend suggests that while some wave feature incorporation is beneficial, overly emphasizing them does not further enhance performance, likely due to the noise introduced at higher values.

**Mooc Dataset:** The AUC remains stable between 0.7035 and 0.7492 regardless of $\alpha$, indicating that the wave feature has a limited impact on anomaly detection in this dataset. This stability implies that Mooc's data distribution may inherently lack high-frequency variations, making wave amplification less useful.

### I.3 CORRELATION ANALYSIS

The correlation between $\alpha$ and AUC also varies across datasets. In the Wikipedia Dataset, there is a weak positive correlation, indicating stable performance as $\alpha$ increases. In contrast, the Reddit Dataset shows a strong positive correlation at lower values, with a rapid rise in AUC when $\alpha$ is between 0.0 and 0.2. The Mooc Dataset exhibits almost no correlation, emphasizing the model's robustness to $\alpha$ changes in this context.

### I.4 CROSS-DATASET COMPARISON

The distinct patterns observed across datasets highlight the importance of contextualizing parameter tuning. While the Wikipedia and Reddit datasets benefit from incorporating wave features, the Mooc dataset demonstrates intrinsic stability, making fine-tuning less critical. The observed differences underline the importance of adaptive tuning strategies when deploying TADNet on diverse data sources.

### I.5 RECOMMENDATIONS

Based on the above analysis, we recommend: 1. Setting $\alpha$ around 0.8 to 1.0 for Wikipedia, as higher values generally improve performance. 2. Fine-tuning $\alpha$ in the range of 0.0 to 0.2 for Reddit to achieve optimal results. 3. Choosing a stable value (e.g., 0.8) for Mooc, as performance remains largely unaffected by variations.

These guidelines ensure that TADNet maintains robust performance across various dynamic graph scenarios, leveraging wave features where they are most beneficial while avoiding overfitting.

## J ADDITIONAL PARAMETER SENSITIVITY ANALYSIS

This appendix presents the complete parameter sensitivity analysis for the proposed model. The analysis evaluates the model's performance across four key metrics: AUPR (Area Under Precision-Recall Curve), F1 Score, and Precision.

### J.1 AUPR ANALYSIS

The Area Under the Precision-Recall Curve (AUPR) evaluates the model's ability to maintain high precision and recall across thresholds. For the Wikipedia and Reddit datasets, AUPR peaks robustly between $\alpha = 0.6$ and 0.7, indicating that moderate emphasis on high-frequency wavelet features effectively enhances the model's detection capability. The performance at $\alpha = 0.9$ is notably lower for these datasets, showing decreased robustness beyond this range. In contrast, the Mooc dataset benefits from a higher $\alpha$ around 0.9, where AUPR reaches its maximum, suggesting that in noisier and more complex environments, stronger reliance on high-frequency components is necessary to improve anomaly detection.

### J.2 F1-SCORE ANALYSIS

The F1-score, which balances precision and recall, reaches its maximum within $\alpha = 0.6$ to 0.7 for Wikipedia and Reddit, further confirming that this range offers the best trade-off between true anomaly detection and false alarm avoidance. For Mooc, the optimal F1-score shifts to $\alpha$ near 0.9,

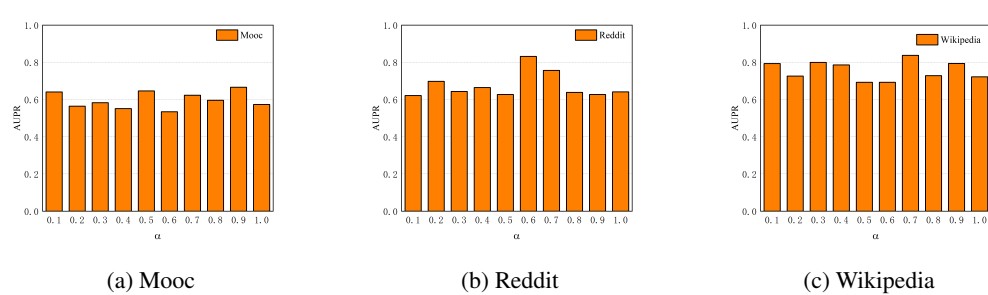

(a) Mooc            (b) Reddit            (c) Wikipedia

Figure 9: Comparison of AUPR values for different $\alpha$ across datasets

consistent with the AUPR results and highlighting dataset-dependent sensitivity. These findings suggest that tuning $\alpha$ based on dataset characteristics is crucial to fully leverage the wavelet-based high-frequency features for anomaly detection.

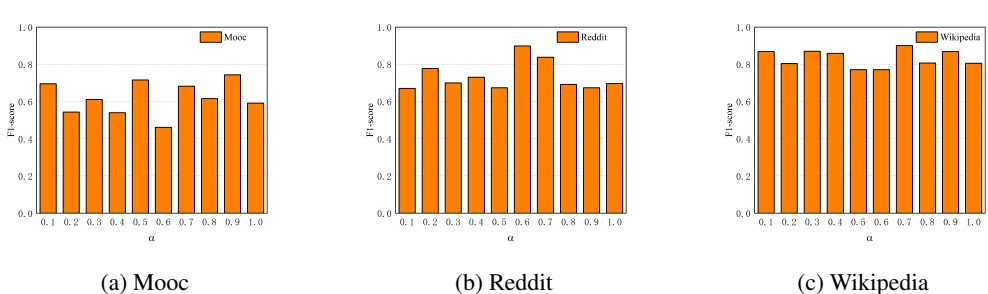

(a) Mooc            (b) Reddit            (c) Wikipedia

Figure 10: Comparison of F1-score values for different $\alpha$ across datasets

## J.3 PRECISION ANALYSIS

Precision measures the proportion of correctly identified anomalies among all positive predictions. For the Wikipedia and Reddit datasets, precision peaks around $\alpha = 0.6 \sim 0.7$, indicating that a moderate weighting of high-frequency details effectively reduces false positives. In contrast, the Mooc dataset achieves its highest precision near $\alpha = 0.9$, reflecting its noisier nature that necessitates stronger emphasis on anomaly-sensitive high-frequency components to improve prediction performance.

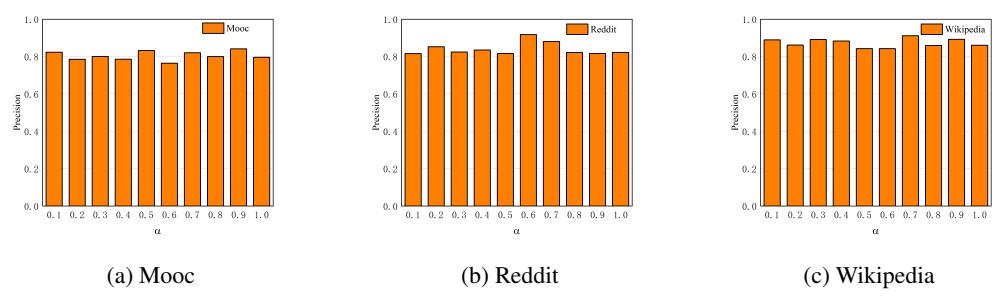

(a) Mooc            (b) Reddit            (c) Wikipedia

Figure 11: Comparison of Precision values for different $\alpha$ across datasets

In summary, the hyperparameter $\alpha$ plays a crucial role in balancing the original node features and the high-frequency wavelet components extracted by the WFP module. Across multiple evaluation metrics, $\alpha$ values in the range of $0.6$ to $0.7$ generally provide the best performance for Wikipedia

and Reddit datasets, while Mooc requires a slightly higher $\alpha$ (around 0.9) to optimize detection results. This analysis highlights the importance of selecting an appropriate $\alpha$ to maximize the model's robustness and generalizability for dynamic network anomaly detection.

## K  LIMITATIONS

While TAD-NET demonstrates strong performance in dynamic graph anomaly detection, several limitations remain. First, the current framework faces scalability challenges when applied to very large-scale graphs, as both memory and computational requirements increase significantly with graph size. Second, although TAD-NET is effective for the anomaly types present in our benchmark datasets, its generalizability to a broader range of anomaly patterns—such as collective, contextual, or evolving anomalies—requires further investigation and potential methodological enhancements. Third, the interpretability of the model's predictions is limited, making it difficult for practitioners to understand the underlying reasons for detected anomalies or to gain insights into the decision process.

To address these limitations, future work will focus on: (1) developing more efficient algorithms and distributed implementations to enable scalability to massive graphs; (2) extending the framework to better capture and distinguish diverse and complex anomaly types; and (3) incorporating explainable AI techniques to enhance the interpretability and transparency of anomaly detection results, thereby facilitating real-time and actionable insights in practical applications.

