# OpenReview forum: "TAD-Net: Reinforced Anomaly Generation and Wavelet-enhanced Prediction for Temporal Anomaly Detection"
_ICLR.cc/2026/Conference — ICLR 2026 Conference Withdrawn Submission_

### Official Review · Reviewer_6XGS · 2025-10-28

**Soundness:** 1
**Presentation:** 1
**Contribution:** 2
**Rating:** 2
**Confidence:** 4

**Summary:**

This paper proposes TAD-NET, a framework for detecting anomalies in dynamic graphs under concept drift. The framework consists of three modules: (1) a temporal feature extractor using graph convolutions, (2) a reinforced anomaly generator combining GANs with reinforcement learning, and (3) a wavelet-enhanced fusion predictor using discrete wavelet transform (DWT). While the paper addresses a relevant problem, it suffers from significant weaknesses in novelty, methodology, theoretical justification, and experimental validation.

**Strengths:**

1. The paper tackles concept drift in dynamic graph anomaly detection, which is a relevant problem

2. The experimental evaluation covers multiple real-world datasets

3. The writing is generally clear and the paper is well-organized

**Weaknesses:**

1. The paper presents a rigid combination of existing techniques without substantial innovation.

2. the "reinforcement learning" component is simply using the discriminator output as a reward signal—a technique already present in MAMF (Hong et al., 2025), which is cited as a baseline. The paper does not clearly articulate what is fundamentally different from MAMF's approach.

3. Wavelet-Enhanced Fusion: DWT for anomaly detection is well-established (Lu & Ghorbani, 2008; Zhou et al., 2020; Liu et al., 2020). The fusion mechanism (Eq. 9) is a trivial weighted sum. EawT (Zhou et al., 2020), MWNet (Shang et al., 2024), and AutoWave (Liu et al., 2020) all use DWT more sophisticatedly for anomaly detection.

4. The core issue is that these three modules address different aspects of the problem in isolation and are not organically integrated. The temporal extractor handles evolving features, the generator addresses sample scarcity, and the wavelet component captures high-frequency signals—but there is no principled framework unifying these components. They appear to be "bolted together" rather than designed as a coherent system.


5. Reinforcement Learning Component:
The claim that this is "reinforcement learning" is overstated. Using the discriminator output as a reward (Eq. 6) without a proper MDP formulation, state transitions, or policy optimization framework is not true RL. This is essentially adversarial training with a reward-weighted objective, which has been done before (e.g., in energy-based GANs and various GAN variants).

6. Assumption 4.1 assumes Lipschitz continuity of normal evolution (‖X_t - X_{t-1}‖_F ≤ L_X), but this is unrealistic in the presence of concept drift. Concept drift by definition violates smooth temporal evolution assumptions.

7. The theoretical analysis assumes a clean decomposition v = s + a (Assumption F.3) where s is smooth and a is anomalous. This is overly simplistic; real anomalies may have both low- and high-frequency components.

8. The paper compares against only 5 baselines, missing several important recent methods.

**Questions:**

See Weakness.

---

### Official Review · Reviewer_3c1x · 2025-10-31

**Soundness:** 3
**Presentation:** 3
**Contribution:** 3
**Rating:** 6
**Confidence:** 4

**Summary:**

The paper proposes TAD-NET, a novel framework for anomaly detection in dynamic graphs that addresses the challenges of concept drift and the scarcity of labeled anomalies. The framework is composed of three components: a temporal feature extractor, a reinforced anomaly generator, and a wavelet-enhanced fusion predictor. These components work together to identify genuine anomalies while mitigating the impact of concept drift by preserving high-frequency features and generating synthetic anomalies.

**Strengths:**

The integration of reinforcement learning into anomaly generation to overcome mode collapse is a significant innovation.

The experimental results show a clear improvement over existing methods, especially in terms of AUC, precision, and F1 score.

The use of temporal feature extraction and wavelet transformations adds technical depth, making the framework robust against evolving data distributions.

**Weaknesses:**

1. While the paper shows that TAD-NET outperforms existing methods, it would be helpful to include a more detailed analysis of specific failure cases where TAD-NET excels compared to the baselines.

2. The method is effective on the datasets used in the experiments, but it would be beneficial to provide more insights into how TAD-NET scales with larger datasets or real-time applications.

**Questions:**

1. Could you provide a more detailed explanation or an example of how TAD-NET handles very high levels of concept drift, especially in large-scale real-world graphs?

2. What is the computational cost of the wavelet-enhanced fusion predictor in terms of time and memory? Would it be feasible to apply this approach in real-time anomaly detection applications?

3. How does TAD-NET perform on data that exhibits highly irregular or unpredictable changes? Are there any cases where the model struggles to distinguish anomalies from natural evolution?

---

### Official Review · Reviewer_naFa · 2025-10-31

**Soundness:** 2
**Presentation:** 3
**Contribution:** 3
**Rating:** 4
**Confidence:** 3

**Summary:**

This paper introduces a framework TAD-NET for anomaly detection in dynamic graphs, with a specific focus on addressing the challenge of concept drift.  The proposed solution is a three-part model: a temporal feature extractor that captures relative changes in node embeddings, a reinforced anomaly generator to synthesize diverse and realistic anomalies, and a wavelet-enhanced fusion predictor to isolate and amplify high-frequency anomaly signals.

**Strengths:**

S1. The authors identify two key problems: the scarcity of labeled anomalies and the difficulty of distinguishing true anomalies from natural network evolution (concept drift).

S2. This paper shows TAD-NET demonstrates significant and consistent performance improvements over baselines across three datasets.

S3. The authors provide theoretical for proposed framework.

**Weaknesses:**

W1. The methodology for the reinforced anomaly generator is a point of significant concern. The generator loss in Equ. 6, L_{G}=-\mathbb{E}_{Z\sim P_{z}}[log~D(G(Z))]+\gamma~log~D(G(Z)) is mathematically ambiguous. The text describes using the discriminator's output probability as a reward in a reinforcement learning framework, but the equation appears to be a simple addition of the standard generator loss (or its negative) and a non-expectation reward term. This formulation is unconventional and confusing.

W2. The ablation study in Section 5.3' analysis is superficial.  What does this "flip" imply about the nature of the datasets or the anomalies within them? Does it mean Mooc's concept drift is more challenging (requiring the RL generator) while Wiki/Reddit's anomalies are more subtle (requiring the wavelet predictor)?

W3. The method combines three existing ideas, the synergy between these specific parts could be argued more strongly. Why is an RL-GAN the right choice for generation over, for instance, a VAE or a diffusion model? Why are wavelets the best choice for separating signals compared to other frequency-domain techniques? Why choose the justification for this specific combination?

**Questions:**

Q1. The authors must rigorously define the loss function in Equ.6. Is it a weighted sum of a standard GAN loss and a policy gradient loss? If so, the policy gradient term is missing its \nabla \log \pi component. If it is a novel hybrid loss, it requires a much more detailed theoretical justification.

Q2. Related to W1 again, the paper is vague about what "removing the reinforcement learning rewards mechanism" means. Does this set \gamma=0$ in Equ. 6? If so, what does the generator loss become? Does it revert to a standard GAN?

---

### Official Review · Reviewer_BLgR · 2025-11-01

**Soundness:** 2
**Presentation:** 2
**Contribution:** 2
**Rating:** 2
**Confidence:** 4

**Summary:**

This manuscript presents TAD-NET for temporal anomaly detection in dynamic graphs. The approach combines GCN-based temporal encoding, GAN-augmented sample generation, and DWT-based feature fusion. While the problem is relevant, the work suffers from inadequate innovation, inconsistent experimental design, and insufficient validation of core claims.

**Strengths:**

1. Addresses concept drift in dynamic graph anomaly detection, a practically important challenge
2. Provides evaluation across multiple real-world datasets with various metrics
3. Includes extensive appendices with theoretical derivations

**Weaknesses:**

1. The framework assembles established techniques without clear innovation. GCN temporal modeling follows JODIE/TGAT conventions, the GAN generation closely resembles MAMF's approach without articulating key differences, and DWT fusion (Eq. 9) is a straightforward weighted sum already explored in AutoWave, EawT, and MWNet. The three modules appear mechanically combined rather than organically integrated to solve a unified problem.

2. The separation of anomalies (high-frequency) from concept drift (low-frequency) lacks empirical validation. No frequency spectrum analysis or ablation study demonstrates this fundamental premise holds across datasets.

3. Section 5.5's t-SNE clustering provides only qualitative visualization without quantitative drift metrics (MMD, KL divergence) or controlled drift experiments.

4. Omits ROLAND (You et al., 2022), AddGraph (Liu et al., 2023), CoLA (Liu et al., 2021)—all recent methods directly addressing concept drift or dynamic anomaly detection.

**Questions:**

1. Can you demonstrate via spectral analysis that anomalies concentrate in high-frequency bands while drift remains low-frequency?
2. How does performance vary when concept drift magnitude increases systematically?
3. What distinguishes your GAN-RL approach from MAMF's adversarial meta-learning beyond terminology?
4. Why not make α learnable rather than a fixed hyperparameter requiring per-dataset tuning?

---

### Note · Authors · 2025-12-05

I have read and agree with the venue's withdrawal policy on behalf of myself and my co-authors.